

# The role of mitophagy in the development of chronic kidney disease

Kexin Yang[1], Ting Li[1], Yingpu Geng[2], Xiangyu Zou[1], Fujun Peng[1] and Wei Gao[1]

[1] Department of Pathophysiology, School of Basic Medical Sciences, Shandong Second Medical University, Weifang, Shandong, China
[2] School of Clinical Medicine, Shandong Second Medical University, Weifang, Shandong, China

## ABSTRACT

Chronic kidney disease (CKD) represents a significant global health concern, with renal fibrosis emerging as a prevalent and ultimate manifestation of this condition. The absence of targeted therapies presents an ongoing and substantial challenge. Accumulating evidence suggests that the integrity and functionality of mitochondria within renal tubular epithelial cells (RTECs) often become compromised during CKD development, playing a pivotal role in the progression of renal fibrosis. Mitophagy, a specific form of autophagy, assumes responsibility for eliminating damaged mitochondria to uphold mitochondrial equilibrium. Dysregulated mitophagy not only correlates with disrupted mitochondrial dynamics but also contributes to the advancement of renal fibrosis in CKD. While numerous studies have examined mitochondrial metabolism, ROS (reactive oxygen species) production, inflammation, and apoptosis in kidney diseases, the precise pathogenic mechanisms underlying mitophagy in CKD remain elusive. The exact mechanisms through which modulating mitophagy mitigates renal fibrosis, as well as its influence on CKD progression and prognosis, have not undergone systematic investigation. The role of mitophagy in AKI has been relatively clear, but the role of mitophagy in CKD is still rare. This article presents a comprehensive review of the current state of research on regulating mitophagy as a potential treatment for CKD. The objective is to provide fresh perspectives, viable strategies, and practical insights into CKD therapy, thereby contributing to the enhancement of human living conditions and patient well-being.

## INTRODUCTION

Kidney disease constitutes a significant global health challenge, incurring substantial economic burdens worldwide in the form of medical care expenses, emergency room visits, treatments, and more. Moreover, kidney-related diseases often give rise to profound mental health issues among patients and their families, owing to the intricate and severe nature of these conditions. Most kidney-related diseases fall into two main categories: acute kidney injury (AKI) and chronic kidney disease (CKD). AKI includes a range of pathological manifestations characterized by rapid loss of kidney function over a short period of time (*Thomas et al., 2015*), often caused by the use of chemotherapy agents such as cisplatin, episodes of renal ischemia/reperfusion, and exposure to contaminants (*Wang et al., 2016*).

Corresponding author
Wei Gao, gaowei@wfmc.edu.cn

AKI is closely linked to elevated rates of morbidity and mortality, and its poor prognosis can lead to renal fibrosis, resulting in the development of CKD. Around the world nearly 9.1% of adults suffer from CKD (*Bikbov et al., 2020*), and the number of global deaths due to CKD increased by 98% in 1990, with the number of cases in Asia increasing from 202.4 million to 441.2 million from 1990 to 2019 (*Aashima et al., 2022*). The incidence of CKD is on the rise globally, with a current prevalence of 13.4%. More than 120 million people in China have CKD, and this number is expected to increase (*Liu et al., 2023a*; *Wang et al., 2023a*). It is calculated that by 2030, nearly 5.4 million patients will require kidney replacement therapy, and by 2040 CKD will be the fifth leading cause of death worldwide (*Liyanage et al., 2015*; *Foreman et al., 2018*).

Renal tubular interstitial fibrosis, is characterized by excessive extracellular matrix deposition, is the end of the most common CKD, in addition also contains RTECs excessive apoptosis, mitochondrial dysfunction and cellular REDOX steady-state characteristics and mechanism of destruction involved (*Guarnieri & Barazzoni, 2011*; *Qi & Yang, 2018*; *Ho & Shirakawa, 2022*). The degree of renal fibrosis is associated with a decline in renal function and determines the prognosis of renal disease. So far, renal fibrosis remains a great challenge. The process of renal fibrosis involves a variety of cellular events, such as RTECs damage, inflammatory cell infiltration, and fibroblast activation (*Liu, 2011*). However, the pathogenesis of renal fibrosis has not been fully elucidated, and the existing intervention methods using renin-angiotensin system inhibitors are not effective in preventing or treating renal fibrosis, resulting in high mortality in patients with advanced CKD. Therefore, the in-depth understanding and further research of CKD to develop an effective treatment, in order to reverse the pathological fibrosis of the kidney has become an urgent need. More and more evidence has shown that mitophagy plays a key role in maintaining the homeostasis of renal cells due to the high content of mitochondria in kidney tissue (*Bhatia & Choi, 2019*). Although the pathway and mechanism of mitophagy have been studied, the role in the pathogenesis of CKD and the regulation of mitophagy in the treatment of CKD need to be studied urgently. Preventing or reversing mitochondrial dysfunction and disordered autophagy in CKD is an unmet clinical need and an urgent research focus. This article reviews the research progress and current status of regulating mitophagy in the treatment of CKD, improving mitochondrial metabolism and CKD disease prognosis, in order to provide effective ideas for clinical treatment.

## Survey methodology

Data were searched from the PubMed (https://pubmed.ncbi.nlm.nih.gov/), CNKI (https://www.cnki.net) databases. The keywords used were as follows: chronic kidney disease, mitophagy, renal fibrosis, mitochondria, PINK1, Parkin, FUNDC1, BNIP3/Nix, cardiolipin, obesity-induced CKD, hyperuricemia-induced CKD, diabetes mellitus-induced CKD, UUO, hyperuricemic nephropathy, diabetes-induced CKD, membranous glomerulonephritis-induced CKD, cisplatin-induced CKD. Search criteria included a time span of the last 10 years, CAS 1 journals, and any type of literature. A total of 265 documents were retrieved and screened for relevance of subject terms, credibility of research methodology and validity of research content, and 103 documents were screened to exclude

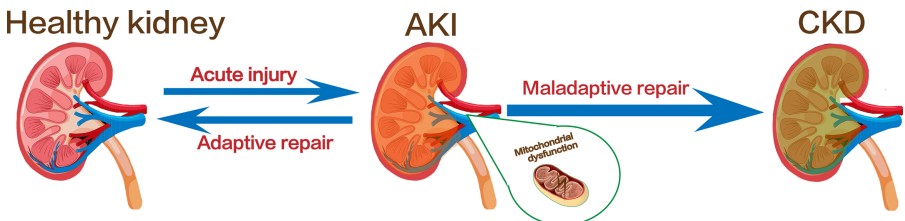

**Figure 1** **Kidney injury and the transformation of AKI to CKD.** Image copyright: FreePik, https://www. freepik.com/free-vector/human-kidney-stones-cartoon-style-infographic_11067983.htm, Premium license.

those with poor quality and duplicate content. We carefully analyzed and summarized the extracted key information, followed by a systematic introduction and in-depth discussion.

## Correlation between CKD and mitochondria

The kidney is a hypermetabolic organ with persistently high energy expenditure. It involves in a variety of important functions to maintain body homeostasis (*Duann & Lin, 2017*). The kidney has the second highest mitochondrial content in the human body, after the heart (*Ralto, Rhee & Parikh, 2020*). Mitochondria is an organelle that produce adenosine triphosphate during various physiological processes (*Nunnari & Suomalainen, 2012*). Under physiological conditions, a balance between mitochondrial fission and fusion and mitophagy together maintain mitochondrial dynamic changes, and the homeostasis of mitochondria is essential for normal cellular function in organs which required large amounts of energy, such as the kidney and heart (*Li et al., 2022*). In addition, mitochondria also ensure their normal morphological structure and function by regulating cell death, inflammation and other cellular processes, and are closely related to kidney injury and repair (*Weinberg, 2011*; *Putti et al., 2015*; *Jin et al., 2022*). Clinical studies have shown that mitochondrial function decline is associated with the degree of kidney fibrosis in CKD (*Bai et al., 2023*; *Jotwani et al., 2024*). In another observational study, a higher mitochondrial DNA copy number was associated with a lower risk of CKD progression, adverse clinical outcomes, and all-cause mortality in patients with CKD (*Liu et al., 2023b*). These evidences all suggest that dysregulation of mitochondrial quality control mechanisms, such as mitochondrial fission, fusion, and autophagy, plays a key role in the progression of renal fibrosis (*Tang et al., 2020a*). Mitochondrial fusion and fission help ensure the maintenance of mitochondrial number and morphology, whereas mitophagy is a form of selective autophagy that degrades and removes damaged or excess mitochondria, controlling the production of mtROS (*Wang et al., 2022*). These three processes interact and act as important components of the mitochondrial quality control mechanisms to regulate the quality and quantity of mitochondria in kidney cells (*Angajala et al., 2018*; *Levine & Kroemer, 2019*; *Dmitrii et al., 2021*). Thus, regulating mitochondrial quality control has emerged as a promising therapeutic strategy to counteract fibrosis progression and prevent the transition from AKI to CKD (Fig. 1).

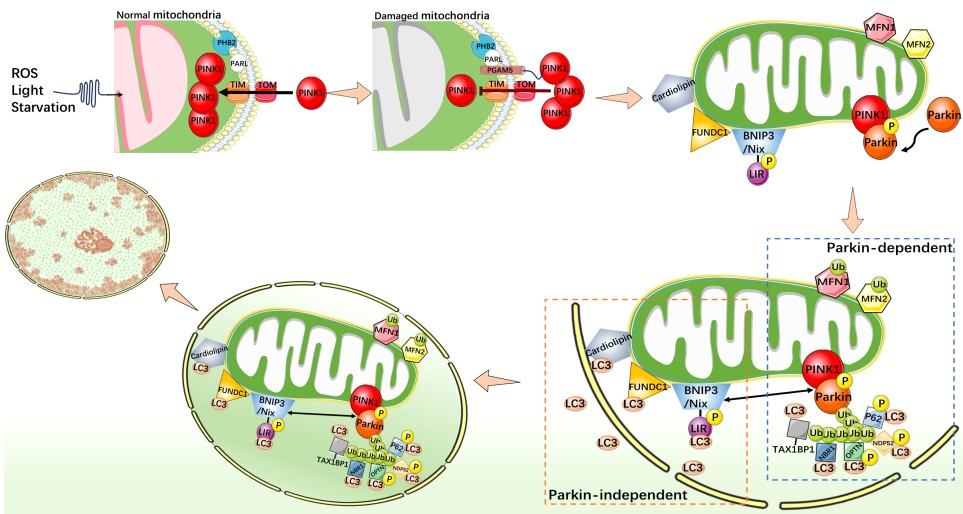

**Figure 2** **Mechanistic diagram of mitophagy.** PINK1/Parkin-dependent mitophagy. Under normal cellular conditions, PINK1 is synthesized and localized to the inner mitochondrial membrane, and enters the inner mitochondrial membrane through the outer membrane translocase and inner membrane translocase (TOM and TIM) complexes. PINK1 is cleaved by MPP and PARL in turn and rapidly degraded in the endosome. When mitochondria are damaged, such as decreased membrane potential or oxidative stress, PHB2 binds to PARL to protect PGAM5. The protected PGAM5 binds to PINK1, and the PINK1 import process is blocked, leading to PINK1 accumulation on the outer mitochondrial membrane and activation of Parkin protein. Activated Parkin selectively ubiquitinated damaged mitochondrial outer membrane proteins such as MFN1/2. P62, TAX1BP1, NDP52, NBR1, and OPTN act as autophagy receptors that bind to ubiquitin chains and interact with LC3. This binding links the ubiquitinated mitochondrial outer membrane to the LC3-I bound phagosome membrane, forming a double-membrane-wrapped phagosome structure, which finally fuses with the lysosome. PINK1/Parkin independent mitophagy. When mitochondria become dysfunctional or damaged, CL, as well as FUNDC1, BNIP3, and Nix undergo specific conformational changes that allow them to expose specific domains that bind to LC3. This binding connects mitochondria to the LC3 bound phagosome membrane and initiates the process of mitophagy.

## Mitophagy

Mitophagy is an indispensable part of the mitochondrial quality control mechanisms, which can remove excess or damaged mitochondria. Under stress conditions, mitophagy is activated as an adaptive or defensive mechanism that can maintain mitochondrial stability. Mitophagy is usually activated by two distinct pathways, one is classical ubiquitin-dependent, such as PINK1/Parkin pathway; another is the receptor-dependent pathway, such as the BNIP3/Nix, FUNDC1, and cardiolipin (CL) pathways (Fig. 2) (*Youle & Narendra, 2011*; *Ashrafi & Schwarz, 2013*; *Chu et al., 2013*; *Scheibye-Knudsen et al., 2015*; *Tang et al., 2020b*).

## PINK1/Parkin pathway of mitophagy

PINK1 is a protein kinase that localizes to the inner mitochondrial membrane (IMM) and is essential for the maintenance of mitochondrial function in cells (*Nguyen, Padman & Lazarou, 2016*). As a key regulator, PINK1 is involved in regulating the health status of mitochondria and maintaining cellular energy supply. It interacts with Parkin (Parkinson protein 2 E3 ubiquitin-protein ligase), an important member of the ubiquitin ligase E3

family, whose function is to link ubiquitin proteins, thus it can target proteins to mark them for degradation. It plays a regulatory role in protein degradation in the cytoplasm. PINK1 and Parkin cooperate closely and play an important role in mitophagy (*Kondapalli et al., 2012*; *Shiba-Fukushima et al., 2012*). In addition, P62 (sequestosome-1) and LC3 (microtubule-associated protein 1A/1B-light chain 3) also play key roles in mitophagy. Through its specific domains, such as the UBA domain and the LC3-binding region, P62 is able to recognize and bind to targeted proteins which were ubiquitinated. However, LC3 exists in two forms, one of which, LC3-I is found in cytoplasm. When mitophagy is initiated and autophagosomes begin to form, the specific enzyme ATG4 processes LC3-I to LC3-II. LC3-II binds to the autophagosome membrane and becomes a key protein during mitophagy (*Stolz, Ernst & Dikic, 2014*). P62 interacts with LC3, especially the LC3-II form, and P62 binds to the autophagosome membrane through its LC3-binding region, facilitating the association of mitochondria with autophagosomes. PHB2 (prohibitin 2) is a highly conserved membrane scaffold protein that participates in mitophagy by interacting with the inner mitochondrial membrane protease PARL. PHB2 regulates PINK1 stability by disrupting the structure of PARL and inhibiting PARL activity (*Yan et al., 2020*). PGAM5, a serine/threonine protein phosphatase that localizes to mitochondria through its N-terminal TM domain, protects the structure of PINK1 and regulates PINK1/Parkin-mediated mitophagy (*Lo & Hannink, 2008*). In the mitochondrial PINK1/Parkin pathway, PHB2 can bind to PARL to protect PGAM5 from cleavage. When the membrane potential is impaired (reduced) and PINK1 is unable to enter the IMM in response to mitochondrial damage, intact PINK1, protected by PGAM5, accumulates in the outer mitochondrial membrane (OMM) and recruits Parkin to mitochondria (*Williams & Ding, 2018*). Then, PINK1 phosphorylates Parkin to activate its E3 ligase activity (*Ding & Yin, 2012*; *Bingol & Sheng, 2016*). Activated Parkin links ubiquitin proteins to their targets on mitochondria, including a variety of mitochondrial proteins such as ubiquitinated MFN1/2. Subsequently, Parkin marks mitochondria in a damaged state. P62 binds to ubiquitinated mitochondrial proteins *via* its UBA domain, recognizing mitochondria as an mitophagy substrate. At the same time, the LC3-binding domain of P62 binds to LC3-II to promote the binding of mitochondria to autophagosomes (*Ding et al., 2010*; *Geisler et al., 2010*). LC3-II further promotes the fusion of autophagosomes with lysosomes to form autophagolysosomes, and removes damaged mitochondria through internal degradation mechanisms to achieve cell purification and health recovery. In addition to P62, TAX1BP1, NDP52 (also known as CALCOCO2), NBR1, and OPTN also act as autophagy receptors in mitophagy. These proteins interact with LC3 family proteins by containing specific sequences or domains, such as the LIR sequence, and bind specific regions on the mitochondrial membrane to LC3 family proteins, thereby promoting selective autophagy of mitochondria (*Wang et al., 2023b*).

## FUNDC1 mediated mitophagy

FUNDC1 protein is a protein present in the outer membrane of mitochondria, combining the typical LC3 motif Y (18) XXL (21) interact with LC3, guide the mitophagy (*Liu et al., 2012*). In addition, FUNDC1 dephosphorylation is essential for the initiation of mitophagy.
Src kinase and CK2 can phosphorylate the tyrosine 18(Y18)13 and serine 13(S13) (*Liu et al., 2012*; *Chen et al., 2014*) sites of FUNDC1 blocking the interaction between FUNDC1 and LC3, thereby inhibiting the occurrence of mitophagy. Meanwhile, PGAM5 dephosphorylates FUNDC1 at S13 (*Chen et al., 2014*). In addition, the mitochondrial outer membrane protein Bcl2-like 1 (Bcl2L1) normally interacts with PGAM5 to inhibit its dephosphorylation of FUNDC1, and depletion of PGAM5 enhances the OPA1-FUNDC1 interaction and decreases the DNM1L-FUNDC1 interaction (*Chen et al., 2016*) thereby inhibiting the activation of mitophagy (*Wu et al., 2014*). However, Bcl2L1 is degraded under hypoxia or mitochondrial uncoupling, leading to the activation of PGAM5, which further dephosphorylates FUNDC1 at S13 and facilitates the interaction between FUNDC1 and LC3 (*Liu et al., 2012*; *Wu et al., 2014*), there by activating mitophagy. FUNDC1 mediated mitophagy also depends on the interactions between the endoplasmic reticulum and mitochondria. Under hypoxia or other stress conditions, FUNDC1 undergoes dephosphorylation, which promotes its binding to calnexin (endoplasmic reticulum) and leads to the accumulation of lipid-rich microdomains on the mitochondrial surface of FUNDC1, which then mediates mitochondrial fission and recruits ULK1 (*Wu et al., 2016b*), thereby activating mitophagy.

## BNIP3/Nix mediated mitophagy

BNIP3L, again say Nix, is a kind of contains only BH3 domain and promote apoptosis of protein structure (*Chen et al., 1999*), mainly locates in the mitochondrial outer membrane, and play a key role in mitochondrial quality control (*Chourasia et al., 2015*; *Lee et al., 2017*; *Tang et al., 2019*; *Fu et al., 2020*). It recruits autophagosomes to mitochondria by directly binding to ATG8-family proteins (*Novak et al., 2010*). The promoter region of Nix gene contains a binding site for hypoxia-inducible factor 1 (HIF-1), which can be upregulated under hypoxic conditions, thereby promoting mitophagy (*Sowter et al., 2001*; *Kubasiak, Bishopric & Webster, 2002*). Another mitochondrial BH3 domain-containing protein is BNIP3 (cl2/adenovirus E1B19-kDa interacting protein 3), which shares 56% amino acid sequence identity with Nix (*Matsushima et al., 1998*). BNIP3 mitophagy activity is regulated by the polymerization of the homologous 2, its protein contains a in C the common domain (coiled coil domain), the domain can make two BNIP3 molecules interact and form stable dimers, enhances the BNIP3 with LC3 interaction ability. This process is related to the presence of a C-terminal transmembrane domain and a WXXL motif (LIR) on the cytoplasmic side of its N-terminus, which binds to LC3 or GABARAP (LC3 homologous protein) in mitochondria to further promote the binding of mitochondria to autophagosomes (*Hanna et al., 2012*; *Zhu et al., 2013*), thereby promoting the selective autophagy process of mitochondria. Maintenance of intracellular mitochondrial mass balance. In addition, BNIP3 also inhibits mTOR activation by sequestering RHEB (*Li et al., 2007*) and activates mitophagy by binding to BCL-2 to block the interaction of the Bcl-2-Beclin1 complex (*Bellot et al., 2009*). In cells to adapt to the low oxygen environment and regulation of cell survival and upstream of BNIP3 molecular HIF-1 alpha plays an important role. Recent study has shown that HIF-1 $\alpha$-BNIP3 mediated mitophagy protects

HK-2 cells from fibrosis by suppressing NLRP3-induced inflammatory responses by reducing the produced mtROS (*Li et al., 2023*).

## CL mediated mitophagy

CL is a protein structure located mainly in the IMM and consists of a central pre-chiral carbon, two phosphates and four acyl chains (*Miranda-Diaz, Cardona-Munoz & Pacheco-Moises, 2019*). As one of the important components of IMM structure, CL participates in the process of mitophagy and apoptosis (*Chu et al., 2013*; *Fritsch et al., 2020*). CL plays an important role in the quality control and morphological regulation of mitochondria, and its function is closely related to its location. When CL is located in the inner membrane, it interacts with OPA1 protein, and this interaction can promote inner membrane fusion to maintain normal mitochondrial structure and function (*Ban et al., 2017*). On the other hand, when CL is located in the outer membrane, it can act as a receptor for LC3 and interact with LC3 protein, thereby participating in the regulation of mitophagy (*Sagar et al., 2023*). CL also can pass in mitochondria PLSCR3 (phospholipid scramblase 3) protein, catalytic CL transfer to the outer membrane, and then combine with LC3. This process could lead to autophagosome to join, and remove the damaged mitochondria, thus promotes mitophagy repair (*Dudek, 2017*). At the same time, CL can also interact with Beclin1, the central receptor of autophagy regulation, to participate in the regulation of mitophagy (*Huang et al., 2012*). In addition, CL plays a key role in regulating the process of apoptosis. It can bind to cytochrome c tightly and unfold part of the protein to form cytochrome c/CL complex, while acting as CL-specific peroxidase to catalyze CL oxidation. Oxidized CL can promote the release of apoptotic factors from mitochondria to the cytoplasm, leading to cell apoptosis. In turn, it can indirectly induce mitophagy by affecting the function and integrity of mitochondria (*Tyurin et al., 2008*).

## Role of mitophagy in CKD

Recent studies have found that mitophagy played an important role in the development of CKD. Through both classical and non-classical pathways, mitophagy can eliminate damaged mitochondria, promote the regulation of energy metabolism, and regulate inflammatory response and apoptosis, thereby reducing oxidative stress and inflammatory response to renal cells and delaying or ameliorating the progression of CKD to reduce kidney injury. Therefore, in-depth understanding and intervention of the mechanism of mitophagy has important clinical guidance and enlightenment for exploring the treatment strategy of CKD (Table 1).

## The role of PINK1/Parkin pathway of mitophagy in CKD

Hyperuricemia nephropathy is a disease caused by the deposition of uric acid in the kidney, which in turn causes inflammation and damage in the renal tubules and interstitium, leading to glomerular damage, and ultimately promoting the development of CKD. The use of fructose-fed CKD model can be used to study the pathological processes such as renal inflammation, oxidative stress and fibrosis caused by high uric acid, so as evaluate the degree of kidney damage. A large number of studies have confirmed that the activation of NLRP3 (NOD-like receptor family pyrin domain containing 3) inflammasome is closely

**Table 1   Mechanisms of mitophagy in CKD.**

| Activation mechanisms of mitophagy | Types of CKD | Mechanism | Role | Animal/Cell model | Literature |
|---|---|---|---|---|---|
| PINK1/Parkin pathway | Obesity-induced CKD | 1. Remove damaged mitochondria 2. Reduce mtROS 3. Inhibition of mitochondrial depolarization 4. Protect mitochondrial function | Activation of mitophagy protects the kidney | HFD-induced C57BL/6 mice, PA-induced HK-2 cell | *Ding et al. (2022)* |
| | Hyperuric acid-induced CKD | 1. Protect mitochondrial function 2. Elimination of mtROS alleviated Sting-NLRP3 axis-mediated inflammation | Activation of mitophagy protects the kidney | Female C57BL/6 mice with fructose-induced uric acid nephropathy and uric acid (UA)—induced hyperuricemic HK-2 cell | *Zhang et al. (2022)* |
| | | 1. Protect mitochondrial function | Activation of mitophagy protects the kidney | Male BALB/c mice were fed with adenine (0.25% adenine) diet and TH1 cells | *Yoon et al. (2021)* |
| | Unilateral ureteral obstruction—induced CKD | 1. Protect mitochondrial function 2. Reduce mtROS | Inhibition of mitophagy protects the kidney | UUO rats | *Jia et al. (2021)* |
| | | 1. Protect mitochondrial function 2. Reduce mtROS | Activation of mitophagy protects the kidney | UUO mice, hypoxia-induced HK-2 cell | *Li et al. (2020)* |
| | | 1. Maintenance of macrophage-mitochondrial homeostasis 2. Reduce mtROS | Activation of mitophagy protects the kidney | Myeloid-specific MFN2-deficient mice, UUO mice, primary human renal macrophages | *Bhatia et al. (2019)* |
| | Unilateral ureteral obstruction and ischemia-reperfusion injury-induced CKD | 1. Impaired mitochondrial function 2. Increased mtROS 3. Promote cell apoptosis | Excessive activation of abnormal mitophagy damages the kidney | UUO mice, IRI mice, TECS (renal tubular epithelial cells) | *Sang et al. (2020)* |
| | Membranous glomerulonephritis-induced CKD | 1. Protect mitochondrial function 2. Reduce the overproduction of mtROS 3. Suppress NLRP3 inflammation | Activation of mitophagy protects the kidney | Membranous glomerulonephritis (MGN) in rats and human podocytes (HPC) | *Wu et al. (2016a)* and *Wu et al. (2016b)* |
| | | 1. Promote macrophage polarization and reduce inflammation 2. Protect mitochondrial function 3. Remove damaged mitochondria 4. Reduce mtROS | Activation of mitophagy protects the kidney | Cationic bovine serum albumin (C-BSA) induced membranous glomerulonephritis (MGN) in rats, human podocytes (HPC), and lipopolysaccharide (LPS) induced mouse macrophage line RAW264.7 | *Cao et al. (2023)* |
| | Cisplatin-induced CKD | 1. Suppress inflammation 2. Protect mitochondrial function 3. Reduce mtROS 4. Restore impaired mitophagy 5. Improve mitochondrial dynamics | Activation of mitophagy protects the kidney | C57BL/6 wild-type and Nrf2 knockout mice | *Ma et al. (2021)* |
| FUNDC1 pathway | Glomerulonephritis-induced CKD | 1. Damage mitochondrial function 2. Increased apoptosis | Excessive activation of abnormal mitophagy damages the kidney | Adenine induced CKD rat model | *Wei et al. (2023)* |

**Table 1** (*continued*)

| Activation mechanisms of mitophagy | Types of CKD | Mechanism | Role | Animal/Cell model | Literature |
|---|---|---|---|---|---|
| BNIP3\BNIP3L/Nix pathway | Unilateral ureteral obstruction-induced CKD | 1. Reduces NLRP3 inflammation 2. The body reaction, reduce inflammation 3. Reduce mtROS 4. Protect mitochondrial function | Activation of mitophagy protects the kidney | HK-2 cells, UUO mice | *Li et al. (2023)* |
| | Hyperuricemia-induced CKD | 1. Promote macrophage polarization and reduce inflammation 2. Protect mitochondrial function 3. Remove damaged mitochondria 4. Reduce mtROS | Activation of mitophagy protects the kidney | Cationic bovine serum albumin (C-BSA) induced membranous glomerulonephritis (MGN) in rats, human podocytes (HPC), and lipopolysaccharide (LPS) induced mouse macrophage line RAW264.7 | *Cao et al. (2023)* |
| Cardiolipin pathway | Ischemia-reperfusion injury-induced CKD | 1. Protect mitochondrial function 2. Remove damaged mitochondria 3. Relieve inflammation | Activation of mitophagy protects the kidney | IRI induced SD rats | *Szeto et al. (2017)* |
| | Obesity-induced CKD | 1. Protect mitochondrial function 2. Remove damaged mitochondria 3. Reduce mtROS | Activation of mitophagy protects the kidney | HFD-induced C57BL/6 mice | *Yeung et al. (2021)* |

related to inflammatory cell aggregation (*Komada & Muruve, 2019*), pro-fibrotic factor release, and extracellular matrix accumulation in CKD. Excessive mtROS can activate the NLRP3 inflammasome and promote the release of pro-inflammatory factors (*Yuk, Silwal & Jo, 2020*), thereby enhancing the inflammatory response. As a key regulator, STING plays an important role in innate immunity and inflammatory response, and it is also a key factor leading to kidney inflammation, injury and fibrosis (*Chung et al., 2019*). Recent studies have shown that STING can affect the activation of NLRP3 inflammasome by regulating mitochondrial function and ROS level. *Zhang et al. (2022)* showed that the expression levels of mitophagy-related proteins PINK1 and Parkin were significantly increased by UroA treatment in a mouse model of fructose-induced hyperuricemic nephropathy (*Ryu et al., 2016*; *Andreux et al., 2019*; *Zhang et al., 2022*). This effect is very beneficial for reducing renal inflammation and inhibiting the activation of STING-NLRP3 pathway. Notably, the inhibitory effect of UroA on the STING-NLRP3 pathway was attenuated when Parkin was silenced. This finding suggested that upregulation of PINK1/Parkin mediated mitophagy can eliminate the STING-NLRP3 pathway-mediated inflammatory response, thereby helping to delay the progression of CKD. MiRNAs play a key role in the repair of renal tissue damage and renal fibrosis. By regulating a variety of target genes, miRNAs have important effects on the response and function of renal epithelial cells and interstitial cells (*Chung & Lan, 2015*). In particular, SIAH3 (Siah E3 Ubiquitin Protein Ligase 3) (*Hasson et al., 2013*), as an E3 ubiquitin protein ligase, can reduce the accumulation of PINK1 in damaged mitochondria, thereby inhibiting the process of mitophagy. However, up-regulated of the upstream miRNA regulator miR-4516 in the renal cortex of hyperuric-induced CKD mice can inhibit the effect of SIAH3, enhance

PINK1/Parkin-mediated mitophagy, reduce dysfunctional mitochondria, and affect the process of cellular fibrosis, thereby improving CKD. This study provides evidence for the importance of miR-4516 in regulating impaired mitophagy in CKD renal fibrosis. Activation of miR-4516/SIAH3/PINK1 pathway by mitophagy can alleviate or reverse mitophagy-deficient CKD, providing a prospective strategy for the treatment of CKD (*Yoon et al., 2021*). Taken together, PINK1-Parkin-mediated mitophagy ameliorated renal fibrosis and alleviated the development of hyperuricemia -induced CKD by reducing mtROS and NLRP3 mediated inflammation.

Obesity-related nephropathy is a kidney disease caused by obesity. Excessive accumulation of fat in glomeruli and renal tubules leads to lipotoxicity of the kidney, which in turn leadings to glomerular damage, abnormal renal structure and function, then develop CKD. The CKD model established by high fat diet (HFD) can be used to study the pathological process of kidney inflammation, oxidative stress and renal tubular injury caused by obesity. Nuclear factor erythroid 2-related factor 2 (Nrf2) is an important REDOX transcription factor. Its activation can enhance the resistance of cells to oxidative stress and reduce mitochondrial damage and functional degradation. It plays a key role in mitochondrial biogenesis and quality control (*Piantadosi et al., 2008*). Recent study has shown that Nrf2 activation can promote PINK1/Parkin mediated mitophagy in obesity-related CKD, effectively reduce oxidative stress level and inhibit mitochondrial depolarization in renal tubular cells, thereby protecting mitochondrial function and reducing kidney damage induced by high fat diet (HFD) (*Ding et al., 2022*). In addition, *Ma et al. (2021)* demonstrated that the use of Nrf2 activator Farrerol could activate Nrf2/PINK1-mediated renal tubular mitophagy and regulate the related oxidative, inflammatory and apoptotic signaling pathways to protect cells from mitochondrial dysfunction and damage. These experimental data suggested that Nrf2 may serve as a novel potential target for the treatment of CKD.

Membranous glomerulonephritis is a chronic progressive glomerular disease, can take advantage of the cationic bovine serum albumin (BSA) C simulation of glomerular filtration membrane damage, to reveal the process of development and relevant pathological mechanism of CKD. *Liu et al. (2022)* used a rat tail vein injection of C-BSA to establish a CKD model and treated it with the mitochondrial-targeted antioxidant MitoTEMPO (*Wu et al., 2016a*). The protein levels of LC3 II/I, PINK1 and Parkin were increased, while the protein level of P62 and mtROS level were decreased in the MitoTEMPO group. This result suggested that MitoTEMPO induces mitophagy in CKD, thereby inhibiting the production of excess mtROS. In addition, we further investigated the role of mitophagy in regulating the NLPR3 inflammasome in podocytes. When Parkin was silenced, NLRP3 expression was significantly up-regulated, suggesting that through PINK1/Parkin-mediated mitophagy, the mitochondrial-targeted antioxidant MitoTEMPO could reduce the production of mtROS, thereby inhibiting the formation of NLRP3 inflammasomes and reducing podocyte injury. In addition, macrophages played an important role in capturing and presenting antigens, secreting inflammatory mediators, regulating proinflammatory and anti-inflammatory responses, and removing cell debris and bacteria. In the progression of CKD, polarization of macrophages is one of its characteristics, which can be divided into pro-inflammatory

(M1) and anti-inflammatory (M2) types. M1-type macrophages lead to decreased renal function by releasing proinflammatory cytokines, which ultimately lead to fibrosis (*Komada et al., 2018*). In contrast, M2 macrophages secrete anti-inflammatory cytokines to repair renal function (*Liang et al., 2021*). Thus, the balance of macrophage polarization plays an important role in the inflammatory state of CKD. Cao et al. used C-BSA to establish a mouse model of CKD (*Cao et al., 2023*). The results showed that the levels of mitophagy-related proteins P62, LC3B, PINK1, Parkin and BNIP3 were significantly decreased in CKD model mice. After using the mitophagy inhibitor (MDIVI), the expression of pro-inflammatory cytokines IL-6 and IL-1 $\beta$ was increased, while the expression of anti-inflammatory factors ARG1 and FIZZ1 was decreased. The trend of macrophage polarization to M1 type was aggravated. However, enhanced mitophagy could reverse the pro-inflammatory effect of MDIVI-1 and significantly alleviate mitochondrial dysfunction. In addition, the role of KLF4 (Kruppel-like factor4) in mitophagy and macrophage polarization was further investigated. After siRNA transfection of RAW264.7 cells to remove KLF4, the expression of mitophagy-related proteins was decreased and the inflammatory response was enhanced. These results suggested that KLF4 overexpression could activate PINK1/Parkin and BNIP3 mediated mitophagy, remove damaged mitochondria, and promote the polarization of macrophages from M1 to M2, thereby improving podocyte injury, immune complex deposition and inflammatory response. It provided a strategy for the prevention and treatment of CKD caused by membranous glomerulonephritis.

Unilateral ureteral obstruction refers to partial or complete obstruction of the ureter, which leads to long-term hydronephrosis of the renal pelvis, which causes oxidative damage and fibrosis of the renal tissue. UUO model is mainly used to study CKD-related inflammatory response, oxidative stress process, tubular atrophy, glomerular sclerosis, interstitial fibrosis and other pathological changes caused by unilateral ureteral obstruction. As mentioned above, macrophages not only play an important role in the inflammatory process, but also play an integral role in the subsequent progression of renal fibrosis. *Bhatia et al. (2019)* used a mouse model of UUO and primary human renal macrophages to demonstrate for the first time that macrophage mitophagy could attenuate renal fibrosis through modulation of the PINK1/MFN2/Parkin pathway. As a mitophagy regulator, MFN2 can promote the entry of Parkin into depolarized mitochondria, thereby inducing mitophagy. When MFN2 is specifically lacking in macrophages, macrophages recombine to the profibrotic/M2 phenotype (*Tanaka et al., 2010*; *Ding & Yin, 2012*), resulting in decreased mitophagy and aggravating the extent of macrophage-derived fibrotic response and renal fibrosis. In addition, in the original generation kidney macrophages knockout PINK1 can lead to mitochondrial quality to drop, this is further evidence that the cause of CKD in UUO, mitophagy to maintain steady, macrophages, and mitochondria to reduce the importance of renal fibrosis. Therefore, enhancing macrophage mitophagy mediated by PINK1/MFN2/Parkin pathway may provide a new idea for the treatment of CKD caused by UUO. In addition, recent studies have found that mitochondrial dynamics was closely related to mitophagy, and it is considered as a way to treat chronic kidney CKD. Mitochondrial dynamics referred to the process of mitochondrial morphology and position changes in cells, mainly including mitochondrial fusion and fission. Mitochondrial

fusion involved fusion-related proteins such as MFN1/2 and optic atrophy 1 (OPA1). MFN1/2 localizes to the OMM and forms a bridge between its GTPase domains rich in transmembrane domains and mitochondrial fusion features to promote mitochondrial fusion. OPA1 is located in the mitochondrial inner membrane and regulates the fusion process of the mitochondrial inner membrane by splicing and modifying proteins to produce different forms. Mitochondrial fission involves fission-related proteins such as DRP1 (*Chung & Lan, 2015*), Fis1, and MFF. DRP1 is a cyclic GTPase that localizes to the mitochondrial membrane and mediates mitochondrial fission by contracting the helix. Fis1 and MFF are located in the mitochondrial outer membrane, interact with DRP1, and assist DRP1 to localize to the mitochondrial surface to promote mitochondrial division. Cleaved mitochondria are more likely to be labeled and degraded and eliminated by mitophagy. RCAN1 (calcineurin 1 regulator) is an important regulatory protein (*Fuentes, Pritchard & Estivill, 1997*; *Ermak et al., 2011*), which plays a role in regulating cell signaling by regulating the activity of calcineurin. In UUO and IRI induced CKD models (*Sang et al., 2020*), up-regulated of RCAN1 leads to mitochondrial dynamic imbalance (up-regulation of DRP1 and MFF and down-regulation of MFN1/2 and OPA1) and impaired PINK1/Parkin-induced mitophagy in UUO model. This suggested that reducing RCAN1 expression in CKD could improve mitochondrial quality control and reduce PINK1/Parkin-mediated mitophagy dysfunction in renal tubules, thereby alleviating renal interstitial fibrosis and programmed cell death of renal tubular cells. In addition, *Li et al. (2020)* found that PINK1 and Parkin levels were increased in a mouse UUO model and in HK-2 cells cultured under hypoxia. However, in the kidneys of PINK1-KO and Parkin-KO mice, RTEC mitochondria were severely lost and damaged, and cytochrome C release was increased. In addition, in HK-2 cells, mitophagy missing PINK1/Parkin mediated that generates mtROS increase and mitochondrial damage occurs, promoted the TGF-$\beta$1 to rise, make the downstream Smad2/3 phosphorylation and activation of transforming growth factor TGF-$\beta$1-Smad2/3 signal transduction, It promoted an epithelial-mesenchymal transition (EMT)-like phenotype, which in turn accelerates the development of renal fibrosis. Similarly, *Jin et al. (2022)* found that in UUO model and TGF-$\beta$1-treated HK-2 cells, the application of the mitophagy activator UMI-77 could inhibit the TGF-$\beta$1-Smad2/3 signaling, the activation of the NF-kappaB pathway and the transcription of pro-inflammatory factors, thereby reducing the inflammatory response and fibrosis process. These findings demonstrated that mitochondria play a key role in the regulation of renal fibrosis.

However, another study found that over-activation of PINK1/Parkin mediated mitophagy in UUO rats resulted in impaired mitochondrial function and aggravated kidney damage, thereby accelerating the progression of CKD (*Jia et al., 2021*). This suggested that although activation of PINK1/Parkin-mediated mitophagy could improve mitochondrial morphology and reduce renal inflammation and fibrosis in various CKD model animals, and played a mainly protective role in CKD, excessive or unbalanced mitophagy may negatively affect the disease. Excessive mitophagy leading to excessive loss of mitochondria and release of damage molecules and mitochondrial DNA, leading to increased cellular stress and inflammatory response. In a study on the liver, it was found that impaired

mitophagy in aging macrophages may lead to mitochondrial damage and the subsequent cytosolic release of mitochondrial DNA, leading to STING activation and induction of proinflammatory responses (*Zhong et al., 2022*). This finding has not been studied in the kidney. Therefore, an in-depth understanding of the function and mechanism of PINK1/ Parkin-mediated mitophagy is of great significance to ensure the proper regulation of autophagy in CKD.

## The role of FUNDC1 mediated mitophagy in CKD

In the study by *Wei et al. (2023)*, it was found that over-activated mitophagy may lead to mitochondrial dysfunction and is closely related to the pathogenesis of CKD. They used a adenine (0.75% w/w) feed in 3 weeks to establish the rat model of CKD, and use the HKL ($C_{18}H_{18}O_2$) for treatment. Western blot results showed that the protein levels of FUNDC1, BNIP3 and Nix in the CKD group were significantly higher than those in the control group, while the protein levels of BNIP3, Nix and FUNDC1 were all decreased after HKL treatment. In addition, HKL also ameliorated the decline of renal function in CKD rats, reduced the expression of renal fibrosis markers Col-IV and $\alpha$-SMA, and alleviated renal tubular lesions and interstitial fibrosis. These results clearly indicated that FUNDC1 and BNIP3/Nix-mediated mitophagy were significantly activated in CKD, and HKL protects the kidney by inhibiting excessive mitophagy in CKD rats. This further demonstrated the double-edged sword effect of mitophagy. Therefore, mitophagy mediated by the FUNDC1 pathway may become a potential method for the treatment of CKD in the future, but how it affects CKD still needs to be further studied.

## The role of BNIP3/Nix mediated mitophagy in CKD

In the study by *Li et al. (2023)* they observed a significant increase in the expression levels of BNIP3 and HIF-1 $\alpha$ in UUO mice and a hypoxia-induced HK-2 cell model, as well as a significant up-regulation of TGF-$\beta$1 and $\alpha$-SMA associated with renal fibrosis, accompanied by the activation of the NLRP3 inflammasome. However, in UUO mice and hypoxia-induced HK-2 cells, knockdown of BNIP3 aggravated mitochondrial structural damage, decreased mitophagy and increased mtROS expression, which further enhanced NLRP3-mediated inflammation and aggravated renal fibrosis after UUO. However, with MitoTempo and the specific NLRP3 inhibitor MCC950, the expression of inflammatory factors was significantly reduced, and the degree of fibrosis was also alleviated. Therefore, a therapeutic strategy by targeting HIF-1 $\alpha$ and BNIP3-mediated mitophagy could alleviate renal fibrosis and delay the progression of CKD. In addition, in 2023 *Qin et al. (2023)* reported that COPT could reduce the expression of mtROS and fibrosis markers, reduced the occurrence of apoptosis, significantly improved AKI, and reduced kidney fibrosis in mice with CKD by inducing BNIP3 mediated mitophagy, while having good biocompatibility. Therefore, COPT may be a promising therapeutic approach to treat AKI and delay its progression to CKD. There are differences between the studies published by *Yao et al. (2022)* and the view of *Li et al. (2023)*. According to their results, although BNIP3 was activated, it didn't appear to ameliorate mitochondrial damage directly through mitophagy. But by inducing Peroxisome proliferator activated receptor-$\alpha$(PPAR-$\alpha$)-BNIP3

pathway to maintain mitochondrial steady-state signal, restrain mtROS and inflammatory reaction (*Pawlak, Lefebvre & Staels, 2015*), reduce the expression of fibrosis markers, so as to relieve the progress of CKD. Taken together, the BNIP3/Nix pathway provides a novel idea to ameliorate mitochondrial damage and reduce fibrosis in renal tubular cells, providing a potential target for the treatment and prevention of CKD. However, for BNIP3/Nix way mechanisms in CKD, still need further research to understand.

### The role of CL mediated mitophagy in CKD

*Szeto et al. (2017)* conducted a 9-month experimental study in which SD rats were subjected to bilateral renal ischemia for 45 min each. The results showed that 9 months after acute ischemic renal showed a continuous endothelial injury, inflammation, fibrosis, and glomerular sclerosis. In addition, in the accumulation of a large number of damaged mitochondria, sertoli cell and mitophagy function obstacle. To further explore the effect of mitochondrial protection on kidney injury, the researchers started to apply the mitochondrial protective agent SS-31 one month after ischemia. The results showed that SS-31 effectively protected mitochondrial integrity in endothelial cells, podocytes and proximal tubule cells by regulating the content of central phospholipids in the mitochondrial inner membrane and inhibiting the peroxidation of CL, and reduced the mitophagy of damaged mitochondria, thereby effectively inhibiting the increase of mtROS. In addition, SS-31 down-regulated the expression levels of inflammatory markers IL-1 $\beta$ and IL-18. Furthermore, this intervention strategy significantly attenuated glomerular sclerosis and interstitial fibrosis, suggesting that SS-31 may attenuate the progression of CKD by interacting with CL to improve mitochondrial quality control and inhibit inflammatory and fibrotic processes. In addition, the study of *Yeung et al. (2021)* found that glucagon-like peptide-1 receptor agonists can promote CL synthetase expression level, increase content of CL, and induce mitophagy process, and then to remove damaged mitochondria (*Dudek, 2017*). Further analysis showed that this strategy was effective in reducing the degree of global lipid accumulation in renal damage induced by high-fat diet. Therefore, CL has broad research prospects and can be used as a potential target for the treatment of kidney diseases. The results provided a new therapeutic option for preventing diabetic kidney disease from progressing to CKD.

## CONCLUSION

Comprehensive studies have consistently highlighted the pivotal role of mitochondrial function in both the onset and progression of CKD. Damaged mitochondria can instigate an excessive production of ROS, which in turn triggers an oxidative stress response, causing further harm to kidney tissues. Furthermore, dysfunctional mitochondria can release signaling molecules that contribute to cellular damage and activate inflammatory responses. Mitophagy, a crucial cellular degradation process, is responsible for eliminating aged, damaged, or dysfunctional mitochondria, thereby ensuring cell health. Recent research suggests that boosting mitophagy could emerge as a novel strategy for treating CKD. Activation of the mitophagy pathway holds the potential to remove aberrant mitochondria, mitigate oxidative stress, and reduce inflammation. Additionally, it may curb renal fibrosis

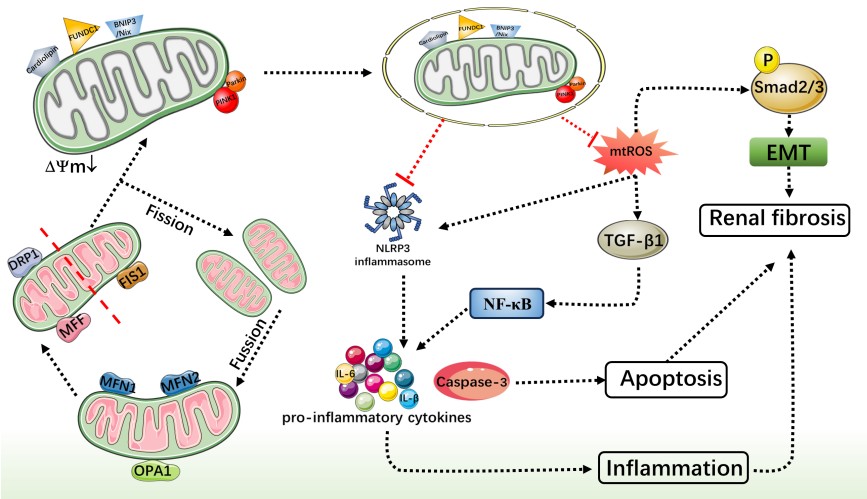

**Figure 3  Protective role of mitophagy in CKD.** DRP1 migrates from the cytoplasm to mitochondria, accumulates around them and cleaves them. Mitochondrial fission separates damaged mitochondria, and excess or damaged mitochondria are eliminated by mitophagy. Mitochondrial damage can cause an increase in mtROS. MtROS increased activation of NLRP3 inflammatory and TGF-$\beta$1's synthesis and secretion, TGF-$\beta$1 through phosphorylation Smad dependence promotes EMT, accelerate the renal fibrosis, and adjust the NF-kappa B pathway and NLRP3 inflammatory corpuscle promote the expression of proinflammatory factor causing kidney inflammation, thus accelerating the process of renal fibrosis. In addition, the activation of NLRP3 inflammasome can promote the expression of caspase-3, cause cell apoptosis, and jointly cause kidney fibrosis.

and safeguard renal tissue from further harm (Fig. 3). This alternative treatment avenue offers patients and their families additional hope and choice during therapy, potentially bolstering patients' mental well-being and morale, thereby aiding overall recovery. Nevertheless, current research regarding specific methods to promote mitophagy for CKD treatment remains in its nascent stages and requires further in-depth investigation to explore mitochondrial therapeutic strategies for CKD. Recently, a new mechanism of mitochondrial fission during mitophagy has been discovered, driven by ATG44 instead of traditional mitochondrial fission factors such as Dnm1/DNM1L/Dpr1. This mechanism promotes mitochondrial fission and subsequently enhances mitophagy, providing a new perspective for treating CKD (*Fukuda et al., 2023*). Therefore a comprehensive understanding of the regulatory mechanisms governing mitophagy and its interactions with other pathological processes is crucial for a more profound comprehension of its role in CKD onset and progression. Although the specific interventions for enhancing mitophagy in CKD treatment are not yet clear, this review lays the foundation for its potential clinical application.

### Funding

This study was supported by the Tai-Shan Scholar Program of Shandong Province (tsqn202103116) and the Program of Scientific and Technological Development of Weifang 2023GX026. The funders had no role in study design, data collection and analysis, decision to publish, or preparation of the manuscript.

### Grant Disclosures

The following grant information was disclosed by the authors:
Tai-Shan Scholar Program of Shandong Province: tsqn202103116.
Program of Scientific and Technological Development of Weifang: 2023GX026.

### Competing Interests

The authors declare there are no competing interests.

### Author Contributions

- Kexin Yang performed the experiments, analyzed the data, prepared figures and/or tables, authored or reviewed drafts of the article, and approved the final draft.
- Ting Li performed the experiments, analyzed the data, prepared figures and/or tables, authored or reviewed drafts of the article, and approved the final draft.
- Yingpu Geng performed the experiments, analyzed the data, prepared figures and/or tables, authored or reviewed drafts of the article, and approved the final draft.
- Xiangyu Zou conceived and designed the experiments, authored or reviewed drafts of the article, and approved the final draft.
- Fujun Peng conceived and designed the experiments, authored or reviewed drafts of the article, and approved the final draft.
- Wei Gao conceived and designed the experiments, authored or reviewed drafts of the article, and approved the final draft.

### Data Availability

This is a literature review.

### Supplemental Information

Supplemental information for this article can be found online at http://dx.doi.org/10.7717/peerj.17260#supplemental-information.

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
