# Peer review of "The role of mitophagy in the development of chronic kidney disease"

_PeerJ, doi:10.7717/peerj.17260_

## Round 0.1 · original submission · Major Revisions

Both reviewers gave suggestions for modification. Authors are requested to revise the manuscript and answer questions.

Reviewer 1 ·

Basic reporting

Yang et al present a complete review about mitochondrial quality control implication in chronic kidney pathology. English is correct and easy to read. Literature references are in general well selected and the use of a summary table helps to follow the article.
In general, introduction adequately introduce the subject, although it would be interesting expand it a little, including some point about other mitochondrial quality control mechanisms, like mitochondrial dynamics, which appear in the rest of the review.

Experimental design

- Methods results excessive short, do not provide enough data of the searching process.
- Although the sections are coherent between introduction and the main part, It would be interesting for authors to create more specific subsections which facilitate reading.
- In lines 115-116, there are other autophagy receptors in addition to p62, like optineurin or NDP52.
- In lines 126-127, it would be interesting include some information about PINK1 processing by PARL.
- In line 136, it should be "autophagolysosomes" instead of "autophagosomes".
- Review lines 168-170, they have no sense.
- In CL sections would be interesting include some explanation or reference about the regulation between mitophagic and apoptotic pathways by lipid peroxidation:10.1016/S0076-6879(08)01419-5
- Lines 358-360 should be explain more in depth or provide some reference. Defective mitophagy induce a mtDNA leakage (10.1111/acel.13622).

Validity of the findings

It would be interesting including some human studies.

Reviewer 2 ·

Basic reporting

I have read the manuscript with interest. An important topic that contributes to knowledge in the area. However, there are several issues that require attention and revision.

Major comments:
1. All the figures presented in the manuscript appear to be pixelated and need to be improved.

2. The following reports are also related to mitochondrial dysfunction in CKD. The authors should include the following references in their manuscript:
• Ho, H.J. and Shirakawa, H. Oxidative Stress and Mitochondrial Dysfunction in Chronic Kidney Disease. Cells. 2022, 12: 88. doi: 10.3390/cells12010088.
• Bhatia, D. and Choi, M. The Emerging Role of Mitophagy in Kidney Diseases. J Life Sci (Westlake Village). 2019, 1:13-22. doi: 10.36069/jols/20191203.

3. A recent report revealed a novel mechanism of mitophagy. Please discuss this issue in this manuscript:
• Fukuda, T., et al. The mitochondrial intermembrane space protein mitofissin drives mitochondrial fission required for mitophagy. Mol Cell. 2023, 83:2045-2058.e9. doi: 10.1016/j.molcel.2023.04.022.

4. Please also put the role of CL in CKD in Table 1. If there is any information about the molecular species composition change of CL in CKD, please include it in this manuscript, because its molecular species composition in different tissues is very important.

Experimental design

It is a review article.

Validity of the findings

no comment

Additional comments

Minor comments:
1. Please correct unnecessary capitalizations.
Line26: Reactive Oxygen Species→reactive oxygen species
Line 36 Keywords: Chronic Kidney Disease→Chronic kidney disease
Line 42-43, Acute Kidney Injury (AKI) and Chronic Kidney Disease (CKD) →acute kidney injury (AKI) and chronic kidney disease (CKD)
Line 96-98: Obesity-induced CKD, Hyperuricemia-induced CKD, Diabetes mellitus-induced CKD, UUO, Hyperuricemic Nephropathy, diabetes-induced CKD, Membranous glomerulonephritis-induced CKD, Cisplatin-Induced CKD→obesity-induced CKD, hyperuricemia-induced CKD, diabetes mellitus-induced CKD, UUO, hyperuricemic nephropathy, diabetes-induced CKD, membranous glomerulonephritis-induced CKD, cisplatin-induced CKD.
Line 107: Another→another
Line 111: inner mitochondrial membrane (IMM), Line 130&192 IMM.
Line273: Membranous Glomerulonephritis (Membranous Glomerulonephritis, MGN) →Membranous glomerulonephritis

2. The words that are shown only once do not need to be abbreviated.
Line64: RASIs
Line180: TMD
Line273: MGN

3. In the author contributions section of this manuscript, some of the authors were involved in conceiving and designing the experiment or analyzing the data. However, it's important to note that this is a review article. Please verify this information.

---

## Round 0.2 · accepted · Accept

Both reviewers gave positive opinions. I reviewed the manuscript and determined that there was no obvious risk of publication and that it was worthy of publication; therefore, I approved the manuscript for publication.

Reviewer 1 ·

Basic reporting

All the proposed changes have been adequately implemented.

Experimental design

All the proposed changes have been adequately implemented.

Validity of the findings

All the proposed changes have been adequately implemented.

Reviewer 2 ·

Basic reporting

The revised article meets the journal's standards. It is now suitable for publication in this journal.

Experimental design

no comment

Validity of the findings

The revised articles complies with the journal's standards.

Additional comments

Well organized article. I look forward to your future developments in this field.